What’s for dinner? Undescribed species of porcini in a commercial packet

Dentinger Bryn T.M. b.dentinger@kew.org
Suz Laura M.
Mycology Section, Jodrell Laboratory, Royal Botanic Gardens , Kew, Richmond, Surrey , UK
Carter Dee
Electronic publication date: 2014 Sep 16
Publication date: 2014
Volume: 2
Electronic Location ID: e570
Received 2014 May 23; Accepted 2014 Aug 20
Copyright: © 2014 Dentinger and Suz
Copyright year: 2014
Copyright holder: Dentinger and Suz
License: This is an open access article distributed under the terms of the Creative Commons Attribution License, which permits unrestricted use, distribution, reproduction and adaptation in any medium and for any purpose provided that it is properly attributed. For attribution, the original author(s), title, publication source (PeerJ) and either DOI or URL of the article must be cited.
License URL: https://creativecommons.org/licenses/by/4.0/

Keywords: Coalescence, Contaminant, Biodiversity, Conservation, Global trade, Naming, Turbo-taxonomy, Phylogenetics, Species

Funding: Royal Botanic Gardens, Kew Funding was provided by the Royal Botanic Gardens, Kew. The funders had no role in study design, data collection and analysis, decision to publish, or preparation of the manuscript.

==============================
Accurate diagnosis of the components of our food and a standard lexicon for clear communication is essential for regulating global food trade and identifying food frauds. Reliable identification of wild collected foods can be particularly difficult, especially when they originate in under-documented regions or belong to poorly known groups such as Fungi. Porcini, one of the most widely traded wild edible mushrooms in the world, are large and conspicuous and they are used as a food both on their own and in processed food products. China is a major exporter of porcini, most of it ending up in Europe. We used DNA-sequencing to identify three species of mushroom contained within a commercial packet of dried Chinese porcini purchased in London. Surprisingly, all three have never been formally described by science and required new scientific names. This demonstrates the ubiquity of unknown fungal diversity even in widely traded commercial food products from one of the most charismatic and least overlooked groups of mushrooms. Our rapid analysis and description makes it possible to reliably identify these species, allowing their harvest to be monitored and their presence tracked in the food chain.

Introduction

Kingdom Fungi is one of the most diverse groups of eukaryotes with estimates ranging from 500,000 to nearly 10 million species, yet they remain vastly underdocumented (Bass & Richards, 2011). Present rates of description, which add on average about 1,200 new species annually (Hibbett et al., 2011), are grossly inadequate for the task. Given that human society has derived tremendous benefit from the foods, medicines, and ecological services provided by as little as 1% of the fungi we know of, the impact of this missing diversity on human livelihoods is potentially profound. Importantly, this missing diversity is not just restricted to remote, underexplored regions of the world, but is a pervasive phenomenon where even our foods can harbor unknown species.

Although taxonomists regard new fungal taxa as commonplace, they are often of little apparent consequence to human society and largely go unnoticed by the public. Like all groups of organisms, our knowledge of fungal diversity is biased towards taxa of greatest concern to ourselves, such as edible fungi. For example, wild mushrooms collected and sold as food around the world generally belong to a handful of well-known taxa (e.g., truffles and chanterelles), most of which have long histories of use in European cuisine. However, even some of these well-known groups have been shown to contain underappreciated levels of diversity. One of these, porcini, has recently been shown to be far more diverse than previously thought (Dentinger et al., 2010; Feng et al., 2012), suggesting the potential for unknown species to end up in the international food supply chain. Although no porcini are known to be poisonous, food allergens have been reported from them (Torricelli, Johansson & Wütrich, 1997; Helbling et al., 2002; Castillo et al., 2013). Therefore, insufficient knowledge of the porcini species contained in food products could pose a health concern.

Porcini are estimated to have an annual worldwide consumption up to 100,000 metric tons (Hall et al., 1998). However, their harvest is restricted to wild foraging since, to date, their cultivation has failed. The high prices for this wild food foraged locally in Europe and North America has driven the market towards less costly sources, such as China (Sitta & Floriani, 2008). According to the official website of Yunnan Province (www.yunnan.cn), the major exporter of wild mushrooms in China, locally-sourced porcini have been exported to Europe since 1973, and mushrooms of Chinese origin now account for approximately half of all dried porcini in Italy (Sitta & Floriani, 2008). The Chinese species of porcini have been shown previously to be more closely related to European Boletus aereus than they are to the core commercial species, B. edulis, with which they last shared a common ancestor up to ∼56 million years ago (Dentinger et al., 2010; Feng et al., 2012).

Given what was previously known about the commercial porcini originating in China, we hypothesized that the contents of a commercially available packet of porcini in the UK would contain multiple species. We set out to rapidly diagnose these species using molecular-based ‘turbo-taxonomy’ (Butcher et al., 2012) that employs a combination of modern tools and approaches. Our results show that, with a combination of phylogenetic taxonomy and e-published nomenclature, three previously unnamed species of porcini could be quickly recognized and formally named from a single packet sold in a London grocer.

Material and Methods

A packet of dried porcini was purchased from a grocer in southwest greater London in October 2013. Fifteen pieces of mushroom were removed arbitrarily from the packet and DNA was extracted using the Sigma Extract-N-Amp kit. The full ITS region of the nrDNA was PCR-amplified using primers ITS1F and ITS4 (White et al., 1990; Gardes & Bruns, 1993). Successful amplicons were purified using ExoSAP-IT (USB, Cleveland, OH) and sequenced bidirectionally using BigDye3.1 with an ABI 3730 (Applied Biosystems, Foster City, CA). Complementary unidirectional reads were aligned and edited using Sequencher 4.2 (GeneCodes, Ann Arbor, MI).

New sequences were combined with 22 related sequences identified using a combination of BLAST searches and the corresponding top hits’ putative species clades reported by Dentinger et al. (2010) and Feng et al. (2012). These related sequences were downloaded from GenBank and correspond to “Boletus sp. nov. 2” (EU231965, EU231966; Dentinger et al., 2010), “Boletus sp. nov. 6” (JN563907, JN563908, JN563909, JN563911, JN563912, JN563913, JN563917; Feng et al., 2012), “Boletus sp. nov. 3” (EU231964; Dentinger et al., 2010), “Boletus sp. nov. 7” (JQ172782, JQ172783, JN563901, JN563902, JN563903, JN563904, JN563905, JN563906; Feng et al., 2012), and “Boletus sp. nov. 5” (JQ563914, JQ563915, JQ563916, JQ563918, JQ563919; Feng et al., 2012). A total of 38 ingroup sequences and one outgroup sequence (Boletus aereus, UDB000940) were aligned using MUSCLE (Edgar, 2004) in SeaView v4.4.0 (Galtier, Gouy & Gautier, 1996) and the terminal gaps converted to missing data. The final matrix consisted of 802 aligned positions, of which 742 were constant and 26 were parsimony uninformative (34 autapomorphic). Minimum and maximum intra- and inter-specific uncorrected “p” distances were calculated using PAUP∗v4.0 (Swofford, 2002). A maximum likelihood tree was generated under a GTR + G substitution model using the Pthreads parallelized version of RAxML v7.0.3 (Stamatakis, 2006; Ott et al., 2007) with nonparametric rapid bootstrapping set to automatically terminate with the ‘autoMRE’ function. A GMYC analysis using the single method (Pons et al., 2006; Fujisawa & Barraclough, 2013) was conducted with the ‘splits’ package (v1.0-18) in R version 2.15.0 (R Core Team, 2014) on an ultrametric tree generated using BEAST v1.8.0 (Drummond et al., 2012). The BEAST analysis applied a rate-smoothing algorithm using an uncorrelated lognormal relaxed clock model (Drummond et al., 2006), the GTR+G substitution model, speciation under a Yule process, the ‘ucld.mean’ prior set to a gamma distribution with a shape of .001 and a scale of 1,000 with all other priors set to default values, and 10 million generations sampling every 1,000 generations. An ultrametric starting tree was provided using the best ML tree from RAxML with branches transformed using non-parametric rate smoothing in TreeEdit v1.0a10. The perl script Burntrees (Nylander JAA, http://www.abc.se/~nylander/burntrees/burntrees.html) was used to sample every 98 trees from the stationary posterior distribution in the BEAST analysis after the first 250 were discarded as the burn-in. These 100 trees were imported for Bayesian GMYC (bGMYC) analysis in R (Reid & Carstens, 2012). Twenty-six GMYC models were evaluated within the 95% confidence and significant clusters were described as new taxa using the ‘turbo-taxonomy’ approach (Butcher et al., 2012), facilitated by the rapid e-publishing tool available through Index Fungorum (www.indexfungorum.org). Voucher material was deposited in the fungarium at the Royal Botanic Gardens, Kew (K) and all sequences were submitted to GenBank (KF815926–KF815937, KF854281, KF854282, KF854283).

Results and Discussion

The GMYC model with the greatest significant ML score included three ML clusters (1–10 clusters with 95% confidence) plus the root (4 ML entities; 2–23 with 95% confidence). GMYC supports for the three ML clusters were weak, low bGMYC posterior probabilities indicated a substantial level of phylogenetic uncertainty, while the maximum likelihood bootstraps supported reciprocal monophyly (79%, 76% and 100% for each cluster respectively; Fig. 1). Percent sequence similarity did not support distinction between any of the three species detected by GMYC and bootstrapping, where the minimum uncorrected pairwise distances between clades was greater than the maximum uncorrected pairwise distances within clades (Table 1). This result suggests that, while GMYC may be particularly sensitive to phylogenetic uncertainty as revealed by the low support values, for this dataset it performs better at diagnosing phylogenetic units than the commonly used percent similarity threshold of 97% (e.g., O’Brien et al., 2005). The phylogenetic uncertainty observed is almost certainly caused by a high ratio of parsimony uninformative variable sites (60 variable positions, 34 parsimony uninformative) to phylogenetically informative changes (26 positions). Of the informative characters, only 11 of them correspond to variable positions between the two closest taxa, B. bainiugan and B. meiweiniuganjun, with five sequences showing heterozygous bases at 6 positions (possibly due to incomplete lineage sorting) and only three of these corresponding to synapomorphic substitutions (Fig. 1). Five sequences contained autapomorphic substitutions in 18 positions, representing more than half of all parsimony uninformative characters, with up to 9 autapomorphies occurring in a single sequence (JN563917). These autapomorphies translate into longer terminal branch lengths relative to internal nodes, which reduces the distinction of within and between cluster branching patterns, a phenomenon that is known to affect GMYC supports (Fujisawa & Barraclough, 2013). These substitutions may indicate true variation in the ITS region, yet 94% come from sequences downloaded from GenBank, with only two sequences (JN563906, JN563917) contributing 83% of the autapomorphies. We suspect that, rather than true variation, these substitutions may instead be the result of sequencing and editing errors. Such errors can have large impacts on phylogenetic inference when the number of phylogenetically informative sites is small, such as in ITS sequences of recently diverged fungi, underscoring the importance of careful scrutiny during sequence preparation.

Figure 1 Phylogeny and alignment of three unnamed species discovered in a commercial packet of dried porcini.

On the left is an ultrametric tree rooted with Boletus aereus and with branch lengths transformed using the uncorrelated relaxed clock model in BEAST. The relationship of the core species or porcini, Boletus edulis, to the dataset is depicted using a dashed line. Clades with dark red branches represent the three maximum likelihood clusters in the GMYC model with the greatest ML score calculated using the single method in the ‘splits’ package in R. Terminal labels in blue represent sequences derived from individual pieces of mushroom sampled from a commercial packet of porcini. Pie charts on branches show maximum likelihood bootstraps (‘MLBS’; lightest red), GMYC supports (‘GMYC’; medium red), and posterior probabilities of the cluster as calculated using bGMYC (‘bGMYC’; darkest red). On the right is the alignment exported from Mesquite v2.75 (Maddison & Maddison, 2011) of 34 variable positions in the ITS region after excluding uninformative sites using PAUP* (Swofford, 2002). Nucleotide characters are depicted using IUPAC codes, gaps depicted by a ‘-’ and ambiguous/missing data depicted by ‘?’.

Table 1 Intra- and inter-specific uncorrected ITS barcode sequence distances of the three unnamed species discovered in a commercial packet of dried porcini.

Ranges are minimum–maximum distances expressed as percent.

	B. bainiugan	B. meiweiniuganjun	B. shiyong	
Boletus bainuigan	0–2.2			
B. meiweiniuganjun	0.4–3.3	0–1.5		
B. shiyong	1.9–4.2	1.5–3.3	0–3.6	

Three species could be identified based on corroboration of ML-supported reciprocal monophyly and GMYC clustering, and these corresponded to lineages previously reported in phylogenetic analyses (Dentinger et al., 2010; Feng et al., 2012; Sitta & Floriani, 2008), but none of which were formally named or described. Review of recent treatments of Chinese boletes also did not provide names for these taxa, which have been treated as a handful of species that occur in Europe and North America (Zang, 2006). New names were formally published on 12 October 2013 (see http://www.indexfungorum.org/Publications/Index%20Fungorum%20no.29.pdf for terse descriptions,1 voucher information, and GenBank accessions corresponding to these taxa). We hope that by naming these taxa and providing reference sequences for comparison, we will encourage mycologists with ready access to fresh collections of these species to record and document their characteristics and discover new features that may help to distinguish them.

Together with improvements in single-locus diagnosis leading to more robust inferences of evolutionary significant units (Butcher et al., 2012), rapid survey and diagnosis of vast communities of undescribed diversity is initiating a revolution in taxonomy (Riedel et al., 2013). This is particularly true for Fungi, which are hyperdiverse and largely cryptic, requiring indirect detection with environmental sequencing for documenting their true diversity (Taylor et al., 2014; Lücking et al., 2014). As a consequence, a vast quantity of fungal diversity is only known from DNA sequences, and these are accumulating in public databases at incredibly rapid rates (Hibbett et al., 2011). Although recent attempts to accelerate species description using short, unique DNA sequences ‘DNA barcoding’ (Hebert et al., 2003) and rapid, short description ‘turbo-taxonomy’ (Butcher et al., 2012) hold promise for meeting the enormous challenge of documenting hyperdiverse and largely unknown groups of organisms (Riedel et al., 2013), they still remain marginal to traditional methods for formal diagnosis of fungal diversity.

Turbo-taxonomy is an important improvement to efficiency in reconciling molecular diagnosis with a standard application of names that enable universal communication about biodiversity. Together, DNA sequence-based diagnosis and turbo-taxonomy catalyze description of new species, thereby greatly accelerating the rate at which diversity can be documented and recognized. Although descriptions based on features of organisms that are readily observed without specialized techniques are ideal, this is not always possible and descriptions based on features of DNA sequences could be automated to satisfy rules on naming. Automated pipelines that integrate analysis, taxonomy, and nomenclature will soon accelerate this revolution, enabling us to capture the most comprehensive baseline information on global organismal diversity possible. Given estimated rates of species extinction from 0.1 to 5% per year (Costello, May & Stork, 2013), and using recent estimates of global fungal diversity of ∼6 million species (Taylor et al., 2014), extinction rates may exceed description rates in Fungi by up to 5 times. An ‘integrative fast track’ approach (Riedel et al., 2013) offers the only tractable solution presently available to filling this knowledge gap. And as has been shown here with the three new species of porcini in a widely available commercial product, this knowledge gap can and does have direct impacts on our lives.

Conclusions

Our analysis of 15 pieces of dried porcini mushrooms from a single commercial packet showed three species corresponding to lineages that although previously reported in phylogenetic analyses have never been formally named or described until now. The recognition of these species enables them to be monitored in foods and facilitates countries’ adherence to international agreements on exploitation of wildlife, i.e., the Convention on Biological Diversity.

Supplemental Information

Supplemental Information 1 Aligned sequences in Phylip format

This is a file containing aligned ITS sequences used for the phylogenetic analyses.

Click here for additional data file.

We are grateful to Rachel Mason Dentinger, who serendipitously supported this research through a spontaneous purchase of dried porcini for our dinner, and to Paul Kirk for nomenclatural advice and for facilitating the e-publication of the taxonomic treatments cited in this study. Meredith Oyen helped locate and translate the Chinese website. We are also grateful for the comments from the editor and four reviewers that improved this report.

Additional Information and Declarations

Competing Interests

Author Contributions

DNA Deposition

1 The numbers reported in the original descriptions should be multiplied by 2.43 to achieve correct measurements of cells and spores.

Bryn T.M. Dentinger is an Academic Editor for PeerJ.

Bryn T.M. Dentinger conceived and designed the experiments, performed the experiments, analyzed the data, contributed reagents/materials/analysis tools, wrote the paper, prepared figures and/or tables, reviewed drafts of the paper.

Laura M. Suz performed the experiments, wrote the paper, reviewed drafts of the paper.

The following information was supplied regarding the deposition of DNA sequences:

GenBank (KF815926–KF815937, KF854281, KF854282, KF854283).

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
