# Peer review of "What’s for dinner? Undescribed species of porcini in a commercial packet"

_PeerJ, doi:10.7717/peerj.570_

## Round 0.1 · original submission · Major Revisions

While the study may be succinct, I do not think it is very easy to understand by the general public and more information about the phylogenetic methods used and how they are interpreted would be helpful. I am confused by the support for branches on figure 1 - an ML score of 76-79 is not normally considered strong and the GMYC supports and probabilities are likewise unconvincing - please explain how these are "proof" of distinct species. As the reviewers note, incorporation of B. edulis in this tree is required, and incorporation of additional porcici species would help.

The title is rather sensationalist since the idea that undescribed mushrooms are present in a packet of edible mushrooms implies that these could be potentially harmful species - these are still species of porcini and as the reviewers note have in fact been described at this stage (and presumably they are not harmful). It would be better to call them "Porcini species other than European P. edulis"

The discussion and conclusions are overstated given that only one small packet (and not even the whole packet!) out of the 10,000 tons of porcini from China have been examined - please tone down the discussion/conclusion to reflect this. Also note comments from reviewer #4 about unsubstantiated claims in the introduction and a need to rework the focus of the paper.

It is my opinion that all of the reviewer comments are fair and valid and all should be addressed in your revised manuscript and accompanying cover letter.

Reviewer 1 ·

Basic reporting

Review of Dentinger & Suz, What’s for dinner?: undescribed species in commercial porcini from China

The authors seem to be using this article on identifying cryptic species of porcini in commercial food products to suggest that the use of DNA sequences is the optimal and fastest method for cataloging fungal biodiversity and describing new taxa. What they refer to as automated diversity diagnosis (sic) is really suggesting that new taxa can or should only be recognized using this technology, which is actually of limited availability to laboratories with the equipment and money for reagents to run the tests. These techniques (molecular biology) do work very well for identification of cryptic, undescribed or difficult to discriminate morphological taxa, but the method should not be portrayed as the panacea for all biodiversity studies.

If they wish to keep the intent and wording of this article, it is suggested they define some of the concepts/terminology: e.g. automated diversity diagnosis – what part of the process is automated? how is the diagnosis made, by comparing DNA sequences? which molecule(s)? what are the criteria for making decisions on taxon identifications – i.e. what percentage of dissimilarity between sequences is acceptable?

It is also suggested they examine the following article which may help with any revisions they decide to make: Riddel et al. 2013 – on integrative taxonomy in Frontiers in Zoology 10: 15 - doi:10.1186/1742-9994-10-15


The authors provide a link to an electronic site with the three new species described and state that these taxa had been recognized in earlier publications as undescribed species. They should at least provide the identifiers of those taxa (the tag used in the phylogenetic trees) so an unambiguous connection is made. Also, the very limited, terse descriptions at the indexfungorum site obviously have some important errors for measurement sizes of basidiospores and other cells which should be corrected at that site since those descriptions serve as the protologues for each of those newly described species. A footnote in this online article about a correction factor for those measurements is not the best way to handle such important errors, since that information seems to be a stand alone online article that will be used for citation of these new taxa.

In the Reference section, Thai should not be italicized, Herbert et al. should come before Hibbett et al.

Experimental design

If they are comparing these taxa to European porcini, then B. edulis and relatives should be in a comparative phylogenetic tree to confirm the uniqueness. If the authors wish to make reference to previous publications, then the actual identifiers in those papers for each new taxon discussed in this paper should be clearly stated.

Validity of the findings

results are valid.

Additional comments

Please consider comments provided in the basic reporting section carefully. Please read the article by Riddel et al. to provide perhaps another approach to your basic premise and to provide readers with a clearer, more precise sense of what you are proposing.

Annotated reviews are not available for download in order to protect the identity of reviewers who chose to remain anonymous.

Reviewer 2 ·

Basic reporting

The manuscript deals with species identification and delimitation of Chinese porcini mushrooms purchased in London market by using ITS sequences. Three species were identified from a single packet of porcini mushrooms. Well written.

Experimental design

Molecular phylogenetic analysis and GMYC analyses were conducted based on ITS sequences to identify and delimit species.
Lines 50-51: “Fifteen pieces of mushroom were removed from the packet and DNA extracted using the Sigma Extract-N-Amp kit.” How did you select the pieces? Randomly? Is there any statistic consideration? Why 15 pieces? Why just a single packet?
Lines 81-86: weak GMYC supports and low bGMYC posterior probabilities indicated a substantial level of phylogenetic uncertainty, while the maximum likelihood bootstraps supported reciprocal monophyly of the three clusters. Is GMYC analysis not a suitable approach to delimit the species of porcini?

Validity of the findings

Some members of the porcini in East Asia were usually regarded as Boletus edulis in its very broad sense in the past in mycological community. The authors have found the mushrooms contained within a commercial packet of dried Chinese porcini purchased in London represent three undescribed species and required new scientific names. In my opinion, the findings are correct and sound, but are not new at all. Some of the findings have been published in Dentinger et al (2010), Feng et al. (2012), and the new taxa have recently been validly published based on molecular and morphological evidence by Dentinger (2013).

Additional comments

This work could make reliably identification of these species in the food chain, but it might not “lead to an improved ability to regulate their harvest and trade, and to monitor potential adverse health effects from their consumption” as stated in the abstract.
Line 91-92 “for details on morphology and voucher information” might be written as “for brief morphology and voucher information”. The descriptions in Index fungorum were not in detail.
The new taxa were published as “Boletus bainiugan Dentinger, in Dentinger & Suz, sp. nov.”, “B. meiweiniugan Dentinger, in Dentinger & Suz, sp. nov” and “B. shiyong Dentinger, in Dentinger & Suz, sp. nov”… suggesting that the names “Boletus bainiugan”, “B. meiweiniugan” and “B. shiyong” were described in the work of Dentinger & Suz by Dentinger. However, taxonomic description of the three taxa in the present manuscript was not provided. Is there any additional manuscript prepared for that? If not, I would like to suggest the authors to add such information in the manuscript and digital images of the basidiomata of the three taxa to the manuscript, which will enable the reader to understand the species concept.
Lines103-106: For many group of fungi, morphological identification of species is often problematic. However, this is not the case for porcini. To characterize and to name porcini, mycologists should try to use both DNA sequences and morphological (and also ecological) features of organisms that are readily observed without specialized techniques. To name fungi on DNA sequences only is sometimes not turbo-taxonomy. Sequence data, however, like morphological and ecological data, is only one aspect of evidence for the taxonomy.

·

Basic reporting

The submission is well written and presented. It is a concise neat story of hidden diversity within a well known, commercially important group of fungi.

Experimental design

No experimental design was required for this study. The materials and methods are perfectly adequate.

Validity of the findings

The results are well presented and their interpretation is interesting and valid

Additional comments

I just have a couple of comments that the authors can consider.
1) Line 32 - production is incorrect. market or consumption may be more appropriate
2) line 40/41 - the significance of this statement will be lost on many readers - consider expanding.
3) l. 107 - consider rewording 'will soon complete this revolution' - this is an ongoing process.
4) I think that many readers would appreciate seeing the 'core species' B. edulis in the tree

Reviewer 4 ·

Basic reporting

The article is grammatically well written. Figures are relevant to the content of the article and are adequately described and labeled.

My primary concern about the basic reporting of the manuscript involves the way in which it is framed. The authors invoke the diversity of Fungi and the need to accelerate species description as a motivating force behind the study, but this isn’t really what the study achieves. Rather, it would make much more sense to focus on the need for rapid identification of species occurring in foods or other products involving biodiversity exploitation. Changing this focus is straightforward, and could be achieved by deleting the first paragraph of the introduction and most of the final paragraph of Results and Discussion. In addition, there are two poorly substantiated or unsubstantiated claims in the third paragraph of the introduction (lines 32-33 and 34-36) that should be addressed. The figure for worldwide porcini production cites a paper by Dentinger et al. (2010) — this paper is a phylogenetic study and I assume it did not calculate an estimate of world productivity — the actual source of this information should be cited. The second claim is that the root cause of import of Chinese porcini is the comparative cost of European material. This appears to be an assumption on the part of the authors, as no source is cited for this statement. How do we know that “demand outpaces supply (from Europe)” does not alone account for the trend? Also, in the second part of the statement, “and an increasing demand from a growing population and the trend in wild foraged foods”, what is it about the “trend in wild foraged foods” that drives importation from China?

Experimental design

My second concern about the manuscript in its current form is the methodology employed by the authors. Given that multi-locus datasets are cited in the paper and the published descriptions on Index Fungorum include morphological data, I see no valid rationale for using a single-locus barcoding approach to species identification in this case. This paper could be a good test case for the use of GMYC methods to identify taxa that have been identified using other data; however, in that case it would be essential to better demonstrate the comparison between the multi-locus datasets cited by the authors and the GMYC results. This comparison may only require a bit more explanation, as the authors already state (line 86-87) that the 3 species found in this study correspond to previously-reported lineages; the authors should more clearly explain how they came to this conclusion. Another thing that needs to be better explained is the taxon sampling employed — it appears that only 3 species plus an outgroup were used; why?However, a bigger problem with the analysis is that the GMYC support values are extremely low; only the ML bootstrap values (I assume that these are from the RAxML rather than the GMYC analysis) appear to provide any reasonable support (I would not consider 79% or 76% bootstrap values as particularly “strong” support as asserted in line 85) for the clades. Therefore, it would seem that the most reasonable conclusion, if I am interpreting the origin of the ML bootstrap values correctly, is that a phylogenetic analysis of the single-locus ITS data yields similar results to a multi-locus approach, and the that GMYC approach failed to recover adequate support for the species distinguished by these phylogenetic analyses; therefore, the best way to analyze these data seems to be to download ITS sequences from Genbank, align them with sequences obtained from the food product, and perform a phylogenetic analysis. While the GMYC analysis is an interesting approach (and, at least in theory, an improved method of single-locus diagnosis), it doesn’t seem to have worked very well in this case — if that’s not the case, then the results should be explained more clearly.

Validity of the findings

My third, and a more minor, concern about the manuscript is the suggested applications of the conclusions of the paper. Will “recognition of these species” actually “enable better regulations to improve food safety”? These species appear to be safe to eat. Will it actually “enable countries to adhere to international agreements on exploitation of wildlife”? These statements seem to be a real stretch regarding the true impact of the paper, and I would recommend toning them down.

In summary, the manuscript generates some very interesting results and would work very well as a straightforward paper that focuses on the core aspect of the study — identifying the species found in this commercial mushroom product. Most of the issues that I have raised in this review revolve around the problem of the paper being framed as something that it is not:
(1) The paper is not a particularly good example of why single-locus taxonomy and “turbo taxonomy” are useful for addressing the biodiversity crisis. With ample morphological and multi-locus genetic data, this is exactly a case that should not rely on shortcut methods.
(2) The paper is not a particularly good example of how GMYC can be used for species delimitation.

What the paper illustrates very well is that commercially available food products contain unknown and even undescribed species, and that DNA barcoding can aid in identifying them. In the case of porcini mushrooms, phylogenetic analysis of ITS sequences appears to be adequate to achieve these identifications. If the authors focused on this aspect of the study, it would result in a logical, solid, and still interesting paper. Linking the study to addressing the biodiversity crisis and demonstrating the application of coalescent methods at first seem like they increase the broader impact of the paper, but in the end these claims are unconvincing and detract from the paper’s believability and overall quality. I strongly encourage the authors to reframe the manuscript so that it focuses on the study’s real strengths.

Additional comments

The authors may wish to consider the naming of sources of the dried mushrooms in lines 47-49. While I personally favor the naming of sources, it seems that other papers on DNA barcoding of commercial products (e.g., sushi) generally tend to omit the names of the restaurants/products sampled. It might be worth making sure that there are no potential legal issues involved with naming these companies in the paper.

One additional point regarding the footnote regarding incorrect size measurements reported in the original descriptions: this information would be much more useful in an amendment to the original descriptions than it is here; I strongly recommend making these changes directly in (or directly linked to) the original descriptions as a benefit to researchers that do not happen to read the footnote in this manuscript.

---

## Round 0.2 · Minor Revisions

Reviewer #4 has re-reviewed your paper and notes that substantial improvements have been made, however additional improvements to the focus of the paper would be helpful in more appropriately framing the paper in the introduction and explaining taxon selection and GMYC analysis. This is a very thoughtful, fair and balanced review and would lead to a significant improvement in the paper. I strongly recommend that you undertake these minor changes in your resubmission.

Reviewer 4 ·

Basic reporting

Dentinger et al. , “What’s for Dinner?”
Review of revised manuscript. Line numbers correspond to the “changes accepted” version of the document.

In my initial review of the manuscript, I highlighted several issues with the scope of reference, methodology, and conclusions drawn by the authors, and pointed out that many of these issues could be addressed by clearer communication of ideas. The revised manuscript is significantly improved, and I commend the authors on taking each point made by 4(!) reviewers under consideration. Most of my comments on the revised manuscript are fairly minor; however, I believe that improvements can still be made in 3 areas: (1) framing of the project in the introduction, (2) explaining taxon selection in the analyses; (3) describing the GMYC results. My comments below appear in the order in which they appear in the manuscript, but the most substantial ones pertain to these three points.

Lines 27-62: While I respect the authors’ prerogative to place their work in the context of their choosing, I still think that the introduction isn’t as effective as it could be. On rereading the manuscript, I now think it’s not the discussion of fungal biodiversity per se, but that the first paragraph gives no indication of the actual scope of the study. After the first paragraph of the study, I would expect the paper to be about a large-scale biodiversity survey of some previously unsurveyed major habitat. What seems to set the introduction on the wrong track for me is how it puts ‘DNA barcoding’ and ‘turbo taxonomy’ at the center, as ways to address the rate of fungal biodiversity description. However, this is not what the paper is about. In my opinion, the introduction could be made much more effective by simply rearranging elements that are already there, and adding a couple of ideas: keep the first two sentences, but then describe how biodiversity impacts us in terms of the products that we use. Then – and this seems to me to be the main point and the main strength of this paper – describe how even our foods harbor unknown biodiversity. After making those points, THEN talk about the methods rather than making them the major focus. Presumably the objective of the study was to find out what was in the packet of porcini, not to develop a test case for the use of coalescent methods for DNA barcode analysis, and with a little more editing of the introduction, I think this point could be very effectively communicated.

Lines 57-58: “millions of years ago” is too vague; please provide the actual range estimated in the cited analysis.

Line 62: a one-sentence statement of the major result would be would give a nice preview to the reader.

Line 65: please correct the phrase “from a in southwest greater London”

Line 67: Recommended change from “DNA extracted using” to “DNA was extracted”

Lines 73-77: I would disagree with some of the other original reviewers, and agree with the authors, regarding taxon selection. I do not believe that the relationship of the species presented in this study to Boletus edulis is relevant enough to the goals of the study to warrant including it in the analysis, especially given the difficulties in sequence alignment (including it in the figure in the way that the authors did seems to me to be a good solution). However, without a description of the method used in selecting taxa, the selection of what WAS included seems arbitrary. The description of the selection of other sequences used in the analysis (lines 73-77) is an improvement, but still needs a little work. The authors now explain that sequences for Boletus spp. nov. 2, 3, 6, 7, and 5 were included in the analysis, but the question still remains as to why. It would be reasonable to assume that a preliminary sequence similarity comparison led the authors to select the taxa that they selected, but I really think that the authors should state explicitly what steps (BLAST search? Phylogenetic analysis?) led them in deciding how to narrow the field of taxa to be included in the final analysis. This explicit description is important for 2 reasons: it helps the reader (and reviewers) to determine whether the taxon selection is valid, and it puts the Yule coalescent method in proper context – as currently described, one might be led to think it a silver bullet that that can bring certainty from chaos; in reality, it is one approach used in concert with other methods, and it seems to be these other methods that guided the authors’ inference of a likely set of taxa to be considered. Again, I do not suggest that the taxon selection be altered in any way, but just that the selection process be more transparently explained.

Line 92: The phrase “smoothing in TreeEdit v1.0a10 on The perl script Burntrees” seems a bit awkward; please check.

Lines 104-122: In my initial review, I concluded that the success of the GMYC approach was overstated in the manuscript. The revised manuscript offers a significantly improved explanation of the results of this analysis; however, two statements provided in this explanation are vague or speculative, and I recommend replacing them with better-supported ones:
• “The phylogenetic uncertainty in this dataset is almost certainly caused by a high ratio of autapomorphic substitutions and insertion/deletion events to phylogenetically informative changes.” This statement is overly speculative, as this quantity can be defined by assessing the data. The authors should strengthen this assertion by providing the exact ratios (or range and summary statistics of ratios across sequences).
• Similarly, the authors should replace the vague statement “These autapomorphies may indicate true variation in the ITS region, although our own observations suggest they may instead be the result of sequencing and editing errors in the sequences downloaded from GenBank, for which we did not have the original trace files to confirm.” Please describe the exact evidence used to come to this conclusion.




References:
Drummond et al. : correct capitalization in the title
Hibbett et al.: “Reviews” misspelled as “Reiews” in journal title
Ott et al. : correct capitalization in the title
Torricelli et al: journal title not italicized


Figure title: “The relationship of Boletus edulis to the dataset is depicted using a dashed line.” May want to change to something like “The relationship of the “true” porcini mushroom, Boletus edulis, to the dataset is depicted using a dashed line,” in order to explain the reason for including B. edulis.


Other points:
In the rebuttal, the authors stated, “However, we hope that by naming these taxa and providing reference sequences for comparison, we will encourage mycologists with ready access to fresh collections of these species to record and document their characteristics and discover new features that may help to distinguish them. Should we say this somewhere in the text?”
I think this would be an excellent idea.

I wish to bring up two additional concerns that have arisen since my initial review. Neither of these affects the scientific validity of the study, but I would ask the authors to consider them anyway. First, I realize that these matters can be out of the hands of authors sometimes, but I found the publicity given on the Kew website to the preprint version of the paper to be in questionable taste, and kind of a thumbing of the nose at the peer review process. It seems best to let the manuscript run its course, consider and/or correct any issues identified by reviewers, and then publicize the study upon publication or at least after final acceptance. Second, the expanded explanation of the additional sequences used in the analyses suggest that the taxa were recognized as both distinct and undescribed in earlier studies by Chinese researchers. The description of the new species on the basis of sequences only, and in a rapid, non-peer-reviewed format, invites the perception (falsely, perhaps) that the Chinese researchers were “scooped.” Ideally, these additional scientists would be included as coauthors of the species in such cases. Neither of these comments in my opinion requires any action or rebuttal from the authors, but I just want to put them forward for consideration.

Experimental design

Experimental design was appropriate to the study and competently performed. Some results could be communicated a bit more explicitly -- please see comments under "Basic reporting."

Validity of the findings

Please see comments under "Basic Reporting."

Additional comments

In general, I find the results interesting and the study appropriately conducted.

---

## Round 0.3 · accepted · Accept

This manuscript has been vastly improved and is now a very interesting and useful piece of work that is set into an appropriate context. Thank you taking the reviewer's suggestions on board to make the requested changes.